# Persistent Droughts and Water Scarcity: Households' Perceptions and Practices in Makhanda, South Africa

Avela Pamla [ID], Gladman Thondhlana *[ID] and Sheunesu Ruwanza [ID]

Department of Environmental Science, Rhodes University, Makhanda 6140, South Africa;
avelapamla@gmail.com (A.P.); s.ruwanza@ru.ac.za (S.R.)
* Correspondence: g.thondhlana@ru.ac.za; Tel.: +27-46-603-7007

**Abstract:** Households in many cities worldwide consume substantial amounts of water, but increasing aridity will result in serious water supply challenges in the future. In South Africa, droughts are now a common phenomenon, with severe implications on water supply for urban households. Developing interventions to minimise the impacts of drought requires understanding of users' perceptions of water scarcity, water use practices, and participation in water conservation practices. Using household surveys across different income groups (low, medium, and high) in Makhanda, South Africa, this study investigates households' perceptions of water scarcity, water use, and conservation practices as a basis for designing pathways for sustainable water use practices. Results indicate that a substantial proportion of households were aware of water scarcity and attributed it to poor municipal planning rather than drought and wasteful use practices. Households reported good water use behaviour, but wasteful practices (e.g., regular flushing of toilets) were evident. Gender, age, education, and environmental awareness influenced water use practices, but the relationships were generally weak. Households participated in water conservation measures but felt the local municipal authority lagged in addressing water supply challenges. The implications of the study are discussed.

**Keywords:** water scarcity; pro-environmental behaviour; water conservation; barriers; interventions

## 1. Introduction

Worldwide, climate change has meant that recurrent, severe, and widespread droughts will become a common phenomenon in the future [1,2], with adverse implications on water supply [3]. In drought-prone areas, persistent droughts can intersect with other drivers of water scarcity, such as population growth, unsustainable consumption and poor management of water, and legacies of inequality and uneven social vulnerabilities, which collectively result in adverse impacts on different sectors including agriculture, energy, and health [3–6]. Meanwhile, projections show climate change will likely increase the frequency and severity of droughts globally, particularly in semi-arid regions [7,8]. Thus, there is a growing concern in society about the socio-economic consequences of droughts and potential interventions for avoiding or minimising drought impacts [7,8]. Among the drought-induced impacts, water scarcity is considered one of the main environmental challenges in both rural and urban settings. However, most scientific research seems to focus more on rural contexts (including farmers' adaptation to drought-induced water insecurity) than in urban contexts. Nevertheless, given that over half of the global population resides in urban areas, and at least two-thirds of the world's population will live in cities by 2050 [9], the role of urban areas in water scarcity debates cannot be glanced over. Among other sectors, the residential sector represents one of the major users of water, consuming approximately 10 billion tonnes of water worldwide [3,10], with these figures likely higher in urban than rural settings. Therefore, the urban residential sector is an important entity in theoretical and practical debates on water security, including development of interventions for addressing the impacts of droughts.

Due to the adverse impacts of persistent droughts, understanding households' perceptions of water scarcity, and water use, and conservation practices can be important for developing contextually relevant intervention strategies to ensure water security. Yet, intervention strategies remain technical-oriented, such as constructing dams, desalination, recycling, and improving water supply [11]. However, a combination of factors including scarce financial and human resources, limited capacity, and insufficient engineering solutions to water resources management means that technical-oriented solutions might be insurmountable, particularly in cash-strapped developing countries [12]. Nevertheless, where technical-oriented solutions are possible, they might be insufficient to address administrative, historical, and behavioural barriers linked to sufficient water supply. Therefore, apart from sheer increases in urban population and the associated rising water demand, future gains from technical-oriented solutions can be eroded if wasteful water use practices are not adequately addressed. Therefore, strategies for responding to drought impacts on water scarcity require a combination of both technical and behavioural responses and lie in understanding people's perceptions and water use practices [4,10,13]. According to Olagunju et al. [14], "awareness of sustainable water use and the subsequent design of appropriate water policies to promote sound water resources management have become key elements of water debates, both in theory and practice, in recent years". Addressing human behaviour as a basis for achieving water security is very crucial, as this is a relatively cheaper, user-driven, and sustainable approach [15].

South Africa is considered a semi-arid country and has been negatively impacted by persistent droughts in the past decades, and lately five provinces have been declared drought disaster zones [16–18]. At the same time, the domestic water sector in the country is grappling with a legacy of inequitable access to water and quality of water services, socially engineered by apartheid era policies of 'separate development based on race' [19]. Meanwhile, the average water consumption in South Africa remains more than the recommended amount needed to sustain water supply [20]. The impacts of persistent droughts in South Africa have especially manifested through strains on water resources in several big cities, e.g., Cape Town and Port Elizabeth (officially renamed Gqeberha), and medium-sized towns, e.g., Makhanda [21–23], where severe water cuts and rationing measures have been implemented to avoid the so called 'day zero' (i.e., the day when most taps in households will be switched off literally). Despite the increasing realisation of the limits of technical interventions in addressing behaviour in general, and water scarcity in particular, the perspectives and water use practices of households in the context of droughts remain little studied. Human perceptions, defined as the process "wherein people select, organise, interpret, retrieve and respond to the information from the world around them" [24], can produce mental expressions and constructions, which can in turn, shape water use behaviour. Understanding households' perceptions of water scarcity, and water use practices can inform drought preparedness plans and intervention strategies for coping with or mitigating drought impacts on water availability. An analysis of perceptions on water scarcity and water use practices across an income gradient is important as it can identify problematic areas and provide insights into development of context-specific interventions.

Within this context, this study aimed to examine urban households' perceptions on water scarcity, water use practices, and conservation strategies, as a basis for informing our understanding of drought impacts on water and crafting potential intervention strategies for dealing with future drought impacts on water. The key questions that guided the study included (i) what are the households' awareness level of water scarcity, (ii) what are the perceived drivers of water scarcity, (iii) what are households' water use practices and responses during water scarcity and what factors influence these, (iv) what are the perceived barriers to implementing water conservation strategies, and (v) what are the implications of the findings on efforts for promoting water security at the household level?

## 2. Materials and Methods

### 2.1. Study Area

Makhanda is a medium-sized town of about 70,000 people, located (33°18′36″ S; 26°31′36″ E) in the Eastern Cape province of South Africa (Figure 1). At an altitude of about 550 m, the town has a warm and temperate climate and is highly susceptible to drought, mirroring evidence of the widespread occurrence of droughts and persistent drying conditions in the country [18]. It has mean temperatures of around 27 °C in summer (October to March) and 19 °C in winter (May to July) months [25]. Mean annual rainfall is about 600 mm per year, with peaks in the summer months October and November and March and April due to frontal rainfall, but rainfall variability and droughts are common. The dominant biome in the town is the sub-tropical thicket, specifically grassland or xeric succulent thicket [26]. Because the town is surrounded by several mountains, patches of South Coast Renosterveld, Afromontane forests, grasslands, and Nama Karoo surround the town [26]. The town is underlain by rocks of the Cape and Karoo supergroup [27].

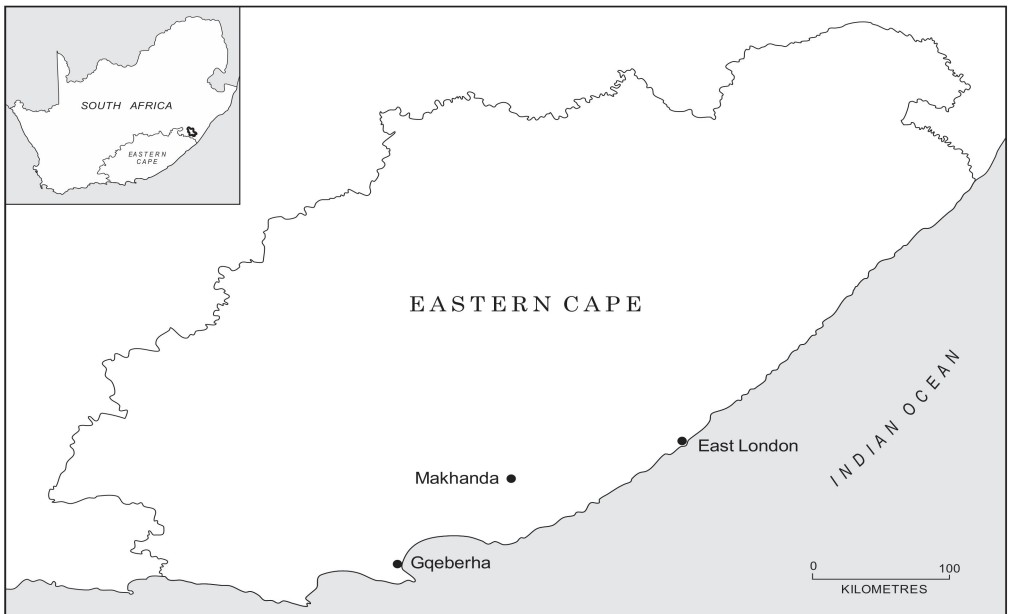

**Figure 1.** Location of Makhanda in the Eastern Cape province of South Africa.

The town was originally established in the 1800s for military purposes during the frontier wars between the Xhosa ethnic group and the European settlers stretching between 1779 and 1879. It is now the administrative town of Makana municipality and is renowned for its contribution to the arts (hosts the Makhanda national arts festival) and education, with several top-tier private and public schools and a university. The spatial structure of the town reflects the colonial racially segregated past, with low- to middle-income households on the eastern side of the town and well-off high-income households on the western side. The west generally has better service provision and looks more affluent than the east, including noticeably cleaner streets, bigger houses and more green spaces, home to Rhodes University, and good private and public schools. Reflecting provincial-level patterns, the town is characterised by low education levels (42% have no or only primary education), limited economic opportunities, a high unemployment rate (34%), and high dependence on government relief grants [28], with these figures higher in the eastern than the western parts of the town. Based on these measures, households in the east are generally more vulnerable to economic and physical shocks such as droughts. Approximately a quarter of the households in Makhanda live below the national poverty line [28].

Two main dams provide the bulk of water needed in Makhanda residential areas. The eastern side gets its water from the James Kleynhans dam and the western side from

Waainek dam. These dams source their water from the Gariep River and the Great Fish River, respectively. The town has faced persistent droughts in recent years with serious adverse impacts and potential threats to the local economy and lifestyles including water supply disruptions [23]. The economy of the town is built around educational institutions, including the local university and private and public schools, which if closed due to water shortages may have negative consequences on the quality of life. These impacts may have disproportionate impacts on vulnerable household groups.

For example, it is not uncommon to find water leaks and treated water gushing out of broken pipes, which has been attributed to old and insufficient infrastructure that has lagged water demand (Water Weekly of 17 May 2019). In light of persistent droughts and water shortages, the municipality recommended different ways for residents to save water, including using water sparingly (50 litres per day per person), switching taps off while brushing teeth, flushing toilets only when it is necessary, limiting laundry to one load a week, reusing towels, using plugs in water basins when rinsing dishes so water can be reused, and taking a five-minute shower instead of bathing. Other technical strategies include drilling of boreholes to access underground water, but this option might not be feasible and sustainable without external funding and skills to drill and maintain the boreholes and upgrading and planned maintenance of existing water infrastructure.

*2.2. Data Collection*

This research was conducted in Makhanda, South Africa between July and August 2019, using structured household surveys. Respondents from two hundred and fifty-one people from households, selected via a stratified random sampling approach, were interviewed using a semi-structured questionnaire. Households were first stratified into three income groups (low, medium, and high income) using location as a proxy for income status [29]. For each income group, the first household was randomly selected (using Google Earth images) and thereafter, every fifth household was selected for interviews. If the targeted fifth household was not available (e.g., no people or adult person), the next available household was approached. We conducted interviews with the head of the household, and in the absence of household heads, adult household members who knew their household characteristics were interviewed. The questionnaire was administered in the preferred language (either IsiXhosa, Afrikaans, or English) and had both closed-ended and open-ended questions.

The first section of the questionnaire captured socio-demographic characteristics of the respondents and their households including gender, age, education levels, household size, employment status, and whether they received social grants or not. In the second section of the questionnaire, the respondents were asked to indicate their main source of household water supply, whether they owned water consumptive equipment or facilities, such as washing machines, swimming pools, cars, water basins, and flush toilets, and whether they knew their daily water consumption. The respondents were also asked to indicate their water use practices via self-reporting. The respondents were presented with a list of water use activities and asked if and how often they practised these activities, with responses selected from a given list on 4-point and 5-point Likert scales. The list of water use activities was informed by and included some municipal-level regulations on water use, including bans on car washing and watering of gardens using hose pipes, and promotion of water-wise actions such as short showers and flushing toilets when needed. For example, respondents were asked how often they flushed toilets and responses included after every use, not much, when I feel it is necessary, and never. In the third section of the questionnaire, the respondents were asked to indicate their level of awareness on water scarcity problems in the town, knowledge of intervention strategies and satisfaction with these, perceived reasons for water supply problems, how this has affected their water consumption levels, and perceived barriers to water conservation.

*2.3. Data Analysis*

All data were captured in an MS Excel spreadsheet for data analyses. Mean scores for each water-use action were calculated from the reported water use practices. Descriptive statistics in the form of proportions, tables, and graphs were used to calculate frequencies of responses including reported water use practices and perceptions of water supply, water use, and water conservation, and effectiveness of water conservation interventions. Parametric tests were used for analysis since the data satisfied normality and homogeneity of variance tests. A one-way ANOVA and Tukey HSD tests were used to find out if there were statistically significant differences in continuous variables between low-, middle-, and high-income groups. Pearson chi-squared tests were used to test for differences in responses between income groups. A Spearman's correlation was performed using Statistica Version 14.0 [30] to explore the relationships between water use practices and socio-demographic factors. Qualitative responses were summarised and recorded, and nominal answers were categorised and assigned numerical scores before analysis.

A potential limitation of this study relates to the subjectivity of measuring behaviour. While self-reporting is a good way of measuring behaviour and can provide insights into actual behaviour, there is a possibility of a yes-saying bias for activities considered as socially acceptable. To address this potential limitation, the purpose of the study was explained to the respondents, and the anonymity of their responses was highlighted before participating in the study, and hence we believe reported practices mirrored what they did in practice. Prior to conducting this study, ethical clearance (Reference Number: 2019-0560-530) was granted by Rhodes University Ethical Standards Committee—Human Ethics subcommittee.

## 3. Results

*3.1. The Socio-Demographic Profile of the Respondents*

Across the sample, there was an overrepresentation of female respondents (73%) in comparison with males (27%). This pattern was reflected in all household groups though high-income households had significantly ($\chi^2 = 251.0$; $p < 0.0001$) more female respondents than low- and middle-income groups (Table 1). The mean age of the respondents was $45 \pm 15$ years, ranging from 19 to 90 years. There were significant (F = 1878.0; $p < 0.0001$) differences in age among low- ($48.4 \pm 16.6$), middle- ($43.8 \pm 15.8$), and high-income ($41.3 \pm 12.2$) groups. The proportion of household respondents falling within the economically active group (19 years and 64 years) was 89%. The average household size was $4 \pm 2$ across the sample with significant (F = 799.9; $p < 0.001$) differences among low- ($3.5 \pm 1.7$), middle- ($4.1 \pm 2.1$), and high-income households ($2.8 \pm 1.5$). Across the sample, about 35% of the respondents had only primary-level education, 28% secondary school-level, and 37% had a tertiary-level qualification. However, low-income households showed a significantly ($\chi^2 = 34.0$; $p < 0.0001$) higher proportion of respondents with primary-level education than middle- and high-income groups, while the latter showed a significantly ($\chi^2 = 55.4$; $p < 0.0001$) higher proportion of respondents with tertiary-level education (72%) compared to the low- (9%) and middle- (29%) income groups (Table 1).

Across all households, 39% received social welfare grants from the state but the proportion was, as expected, significantly ($\chi^2 = 22.7$; $p < 0.0001$) higher for low-income (67%) than for the middle- (43%), and high-income (6%) households. Almost all households interviewed had access to a flush toilet, and overall, more than three-quarters owned a washing machine, although less than half of those with washing machines were from low-income households. Just above half of the households owned a car, but relatively few households owned dish washers and double water basins. Analysis by household group shows clear asset ownership patterns with a higher proportion of high-income households owning household assets including washing machines, dish washers, and cars than middle- and low-income households, which confirms documented disparities in income status by location in South Africa. The different asset ownership patterns suggest that well-off households should be key target groups in water scarcity debates because they are likely to

consume more water than their low- and middle-income counterparts. Yet, the impacts of water scarcity are likely to be felt more by low- and middle-income households due to their lack of safety nets.

**Table 1.** Socio-demographic profile of respondents and household facilities.

| Aspect | Household Group | | | Total Sample (*n* = 251) |
|---|---|---|---|---|
| | Low-Income (*n* = 100) | Middle-Income (*n* = 100) | High-Income (*n* =51) | |
| Gender of respondents (%) | | | | |
| Female | 70 | 69 | 80 | 73 |
| Male | 30 | 31 | 20 | 27 |
| Mean age | 48 ± 16 [a] | 44 ± 16 [b] | 41 ± 12 [b] | 45 ± 15 |
| Mean household size | 3 ± 2 [a] | 4 ± 2 [b] | 3 ± 2 [a] | 4 ± 2 |
| % Households receiving social grants | 67 | 43 | 6 | 39 |
| Education level (%) | | | | |
| Primary | 66 | 39 | 0 | 35 |
| Secondary | 25 | 32 | 28 | 28 |
| Tertiary | 9 | 29 | 72 | 37 |
| Household assets/facilities | | | | |
| Flush toilet | 97 | 96 | 100 | 98 |
| Washing machine | 43 | 87 | 100 | 77 |
| Shower | 12 | 74 | 100 | 62 |
| Car | 22 | 56 | 90 | 56 |
| Garden | 23 | 40 | 78 | 47 |
| Double water basin | 3 | 21 | 69 | 31 |
| Dish washer | 3 | 9 | 31 | 14 |

Means with different letter superscripts indicate significant differences among income groups at *p* < 0.05.

### 3.2. Sources and Consumption of Water

Nearly all households (96%) reported indoor taps as the main source of water supply but out of this only 38% used it for drinking. Analysis by income group showed that a significantly ($\chi^2$ = 64.3; *p* < 0.0001) higher proportion (72%) of low-income households used tap water for drinking compared to 34% and 8% for the middle- and high-income households, respectively. A significantly ($\chi^2$ = 6.8; *p* < 0.009) larger proportion (65%) of high-income households used bottled water as their main source of drinking water compared to middle- (52%) and low-income (21%) households. Other sources of drinking water mentioned by the respondents included spring and rainwater. Concerning water consumption, only 40% of the respondents reported using between 16 and 50 litres per day in line with municipal water restrictions. More than half (55%) of the respondents were not aware of the amount of water they used per day and the remaining 5% used more than 50 litres of water per day.

### 3.3. Perceptions of Water Scarcity

Approximately 93% of all the sample respondents reported that they were aware of the current water scarcity in Makhanda through either personal observations and experiences (32%), social media platforms (12%), radio (10%), television (10%), local newspaper (9%), word of mouth (7%), and awareness flyers (7%). About 62% of the respondents across the sample perceived the supply of water as bad, but significantly ($\chi^2$ = 53.5; *p* < 0.0001) more low-income (85%) than the middle- (65%) and high-income households (35%) felt so (Figure 2).

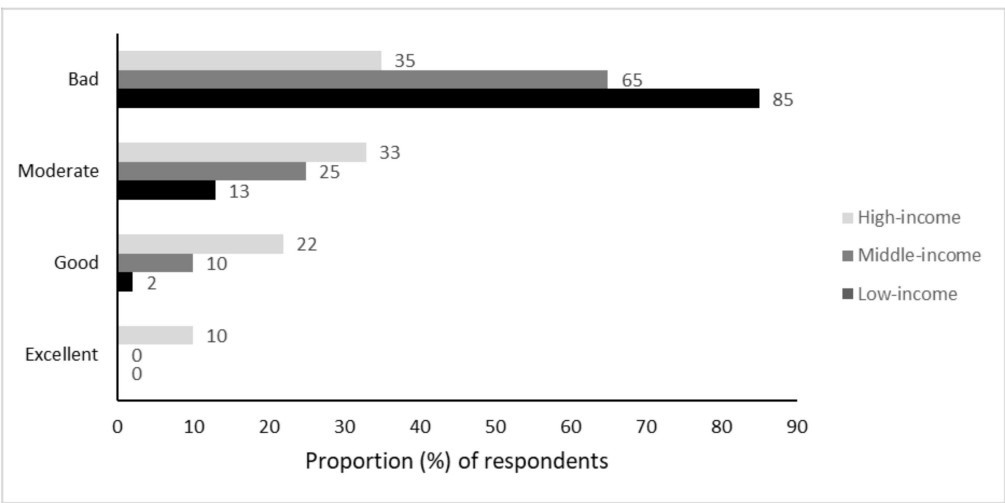

**Figure 2.** Responses to water supply condition in Makhanda.

Overall, a high proportion of low- and middle-income households perceived water supply as bad compared to high-income households. When the respondents were asked to indicate what or who they thought was responsible for the water scarcity problems, more than half (53%) cited poor planning by the local municipality. About 26% attributed the water scarcity problem to recurrent droughts, and 23% felt it was a combination of social, natural, technical, and behavioural factors, increasing population, wasteful water use practices, and ageing infrastructure.

*3.4. Reported Water Use Practices*

The respondents were asked to indicate their water use practices relating to toilet use, doing laundry, car washing, and taking a shower (Table 2). Just about two-thirds (67%) of all the respondents said they flushed the toilet 'when it is necessary'. The remaining proportion flushed 'after every use' (27%) and 'not much' (6%) (Table 2). Analyses by household group showed that a sizeable proportion of households (>60%) across all household groups flushed the toilet when it was necessary, though more high-income households than low- and middle-income households reported so (Table 2).

About 52% of the respondents who owned cars said they washed their cars 'when it is dirty', 27% 'never' washed their cars at home but did so at car wash facilities, and 11% reported washing their cars 'once a month'. The remaining respondents (10%) washed their cars either 'weekly' or 'twice' a month (Table 2). Concerning laundry, a substantial proportion (94%) of the respondents said they did their laundry either 'weekly' or 'bi-weekly' and the rest of the households (4%) did their laundry daily or were not sure (2%). Across the sample, about 61% reported taking showers in five minutes or less in line with current municipal water-saving recommendations, but these figures were more pronounced for low-income (80%) and high-income (67%) households than middle-income (38%) households (Table 2). The remaining proportion reported taking between 6 to 10 min (32%), more than 10 min (4%), and (3%) did not know how long showering lasts.

The results of a Spearman's correlation show significantly ($p < 0.05$) negative but weak to moderate associations between age and the frequency of flushing toilets, and between education level and duration of showers (Table 3). Significant ($p < 0.001$) but weak positive relationships were found between gender and laundry frequency, and between household size and laundry frequency (Table 3).

**Table 2.** Water use practices by household group.

| Water Use Practice | Low Income (*n* = 100) | Middle Income (*n* = 100) | High Income | Sample Mean (*n* = 251) |
|---|---|---|---|---|
| | | | (*n* = 51) | |
| Flushing (*n* = 250) | | | | |
| After every use | 32 | 36 | 12 | 27 |
| Not much | 2 | 4 | 12 | 6 |
| When I feel it is necessary | 66 | 60 | 76 | 67 |
| Never | 1 | 1 | - | - |
| Car wash (*n* = 124) | | | | |
| Weekly | 4 | 2 | 5 | 3 |
| Twice | 14 | 7 | - | 7 |
| Once | - | 18 | 15 | 11 |
| When I feel it is dirty | 68 | 59 | 28 | 52 |
| Never | 14 | 14 | 52 | 27 |
| Laundry frequency (*n* = 251) | | | | |
| I am not sure | 4 | 3 | 3 | 2 |
| Everyday | 10 | 9 | 8 | 4 |
| Weekly | 56 | 75 | 70 | 80 |
| Fortnightly | 30 | 14 | 19 | 14 |
| Shower length (*n* = 72) | | | | |
| ≤5 min | 80 | 38 | 67 | 61 |
| 6–10 min | 20 | 46 | 29 | 32 |
| >10 min | - | 8 | 4 | 4 |
| I don't know | - | 8 | - | 3 |

**Table 3.** Relationship between water use practices and socio-demographic variables.

| Variables | *n* | Spearman (Rho) | *p*-Value |
|---|---|---|---|
| Age and flushing toilet | 250 | −0.136 | 0.031 |
| Gender and laundry frequency | 251 | 0.170 | 0.007 |
| Education level and duration of shower | 72 | −0.436 | 0.000 |
| Household size and laundry frequency | 251 | 0.234 | 0.000 |

*3.5. Water Conservation Measures and Challenges*

In response to prevalent droughts and water scarcity, the respondents said they actively engaged in water conservation measures including water reuse (71%), reduced water consumption for daily household activities such as cooking, laundry, and bathing (26%), storage of municipal water, ensuring that taps are tightly closed (14%), and rainwater harvesting (11%). Households stored water in plastic bottles, buckets, used drums, refuse bins, and water tanks in response to water shortages. Disaggregation by household group showed that more than two-thirds (67%) of high-income households used water tanks for rainwater harvesting. When asked if they knew any interventions for reducing water consumption by the local Makana municipality, only 28% stated that they were aware of the interventions. Concerning challenges to water conservation, nearly half (47%) of the respondents cited lack of awareness, while others said it was not "possible to save water when there was no water coming out of the tap" (21%). Some respondents felt it was just hard to save water since they do not remember water rationing times (20%) and 16% indicated that they did not feel challenged to conserve water. Other challenges mentioned

include unavailability of water tanks for rainwater harvesting (14%), leaking pipes (12%), and hygiene consideration (4%).

## 4. Discussion

The study considered household water use practices and responses to water scarcity in the context of persistent drought-induced water shortages in a medium-sized town. All the survey households had access to tap water. As expected, most respondents were aware of water scarcity problems and rated the supply of water in the town as bad due to frequent and long water cuts. However, notably more low-income households rated the water supply situation as bad than other household groups. This can be attributed to the fact that water cuts are more frequent in low-income areas than well-off areas owing to small reservoirs and continually low reservoir levels supplying these areas. Reservoirs in low-income areas were historically meant to supply water to few households but with an increasing proportion of the urban poor population combined with poor rainfall patterns being experienced in the study area, the reservoirs do not have the capacity to meet rising water demand. Low-income households also have fewer alternative sources of water, which makes water cuts a huge lifestyle disruption. The study lends support to the literature on the disproportionate impact of water scarcity on low-income households, mirroring existing patterns of inequities in resource access and quality of services in urban spaces based on apartheid era spatial and economic segregation [5,19].

The study also shows that despite the general acknowledgement of persistent droughts, more than half of the respondents attributed water scarcity to technical factors, including failure by the municipality to invest in maintenance and expansion of existing water infrastructure in the face of a rising population and water demand, and to attend to water leaks, and a generally poor service provision. The study also notes that less than half of the surveyed respondents complied with the municipal recommended daily water consumption of 50 litres per day. In our case, there is a long-held perception of the local municipality's failure to provide or maintain services, including waste management, roads, and quality water provision [23]. For example, perceptions on poor water quality are deeply held in the town, with well-off individuals opting for spring and bottled water. In our case, more well-off households than poor households used bottled water as a source of drinking water, while more than half of poor households relied on tap water for drinking. This view (municipality's failure) has important implications for engendering a shared responsibility in addressing short- and long-term water supply problems in urban areas, particularly residents' involvement in such interventions. For example, residents might be reluctant to engage in water-saving measures since they may feel it is the municipality's responsibility to 'clean up its incompetence-driven mess'. It is plausible to argue that these perceptions might explain the lack of motivation by a sizeable proportion of households to engage in water-saving measures.

The study also notes evidence of both wasteful and water-saving practices among households. Water-saving practices included flushing the toilet when it is necessary (rather than after every use) and doing laundry weekly. Responses to water scarcity included reducing daily consumption of water in various household activities, and water storage among other water-saving measures. Roseth [31] argues that water conservation behaviour is mainly driven by the motivation of households not to run out of water and does not necessarily indicate environmentally friendly behaviour. The findings suggest that most of the respondents were not aware of the recommendations put forward by the municipality to reduce water consumption. This implies that water conservation is generally viewed as using less water than consciously engaging in specific behaviours to reduce consumption [32]. Other respondents cited forgetfulness and feeling unchallenged as reasons for not implementing water-saving actions. These reasons speak to the influence of personal dispositional factors in water use practices, which has been documented in other sectors, including energy and recycling [33]. Lack of motivation might be rooted in residents' perceptions that the water scarcity problem is external and feelings of lack

of control of the situation—factors which have been found to erode the motivation to act pro-environmentally [34]. Unfavourable contextual conditions, such as the perceived municipality's failure to render services (discussed earlier) and the perceived low efficacy to change the situation, might constrain the propensity to implement water-saving actions.

This implies that there is a need to have platforms for municipal authorities and residents to engage on water issues, and address capacity development issues that can go a long way in addressing technical aspects such as water leaks, burst pipes, ensuring good water quality, and sufficient production of water. Changing residents' perceptions of the municipality's role in the water crisis will require platforms for sharing perspectives. However, given the diversity of household groups, varied perceptions of water scarcity and how to address them are highly likely. Hence, platforms that allow all household groups to be represented might allow consideration of multiple views and voices, development of collective responsibility, and acceptability of measures for addressing water scarcity in the context of persistent droughts. Taken together, the study points to the complex intersectionality of drought and administrative failures and the seemingly disproportionate impact on low-income groups, which should be well understood to develop meaningful strategies for saving water.

Concerning the influence of socio-demographic factors on water-saving practices, the findings suggest females are less likely to participate in water-saving measures than males. This does not necessarily mean females are less environmentally friendly than males. Rather, we argue it shows the direct involvement of females in water-linked household chores, such as doing laundry, cleaning the house, and cooking. For example, in our case, females reported a higher laundry frequency than males. This also highlights the burden that women may endure in the face of water scarcity and hence gendered perspectives to addressing water scarcity are needed. Age showed a negative relationship with the frequency of flushing toilets, suggesting that older people are less likely to flush toilets after every use than younger people. Gregory and Di Leo [35] report that older people consume less water than younger people. In our case, this might be explained by the responsibility older members of households have in household water provision, and negative consequences they might incur for not saving water. The positive relationship between household size and laundry frequency suggests that bigger households are likely to do laundry more often than smaller households. Given that big households are more likely to have a higher number of children than young households, the need for doing laundry regularly is plausible, and by extension high consumption of water, consistent with findings elsewhere [36–38]. The negative relationship between education and shower duration suggests that highly educated individuals are likely to spend less time taking a shower, perhaps owing to their high level of awareness regarding the impacts of drought and wasteful practices on water availability. High levels of education are often associated with environmentally friendly behaviour because highly educated people are likely to be informed about impacts of their actions on the natural environment [39]. It could also be argued that highly educated people are likely to have access to information, e.g., social media platforms where issues around water crises are discussed and communicated. In our case, it is possible that many low-income households are not on social media or other platforms, such as Facebook, residents' associations, and mailing lists, where information on water supply levels and planned water cuts is shared. This means that the likelihood of disruption to social life is higher for low-income households who cannot prepare in advance for water shutdowns.

Concerning barriers to water conservation, more than half of the surveyed households said they were unaware of municipal-level interventions for addressing water scarcity, highlighting the need to raise awareness and share water-saving information among residents. Raising awareness through educating or informing people about the benefits of water conservation has the potential to promote water-saving actions. Purcell and Magette [40] found a positive association between involvement of people in educational programmes and positive behavioural outcomes. Awareness campaigns on household-based measures

for saving water, including doing laundry and flushing toilets when it is only necessary, water reuse (for activities such as washing dishes, washing cars, and watering gardens), and taking short showers, might assist in minimising rising water demand. While the immediate benefits might not be significant in economic terms to households, small water savings across all households might result in a substantial reduction of water demand and minimise water cuts.

The results have practical implications for a reliable availability of sufficient water needed to sustain society. For example, predictions of severe and widespread droughts in the future mean that water security will be a serious service delivery challenge, particularly in contexts where water supply is already problematic [8]. Protecting society from the extremes of drought requires investments in both technical and behavioural strategies for managing water supply in cities. The way water is managed and used will have implications on the supply of water, and in turn, on the social and economic activities needed for human well-being. For example, the local municipality reports failure to supply water to consumers in an equitable way or transfer water to low-level reservoirs due to residents' failure to use water sparingly and reasonably. For example, when water is restored for a few hours, some residents quickly irrigate gardens and fill up their rain tanks and swimming pools, which means it is not possible to transfer water from one area to the other, illustrating the important role of sustainable water use practices in crafting interventions for water security. Beyond the local level, our results have national implications. At the national level, water insecurity is increasingly a huge challenge with many cities reporting water supply failures. While water supply failure is in part explained by climate change-related events (frequent droughts, heat waves, and late onset of rains), equitable use of the little water available will depend on consumers' use practices. While the water supply problem can vary from place to place, it is plausible to suggest that technical intervention measures supported with behavioural strategies can help mitigate the impacts of droughts. A shift from technical-centred interventions to techno-behavioural interventions might ensure supply of reliable and adequate water to residents in the face of increasingly arid conditions, and their negative impacts on water availability. This shift is transformative in its nature because it can allow equitable sharing of scarce water resources, an aspect that is especially important to consider in contexts characterised by huge inequalities in basic service provision, such as water supply. Interventions that build on engendering sustainable water use practices can allow city authorities and consumers to have a shared understanding of the water supply challenge, and highlight the responsibility of each stakeholder in managing whatever water is available.

## 5. Conclusions

In conclusion, understanding household water use practices and responses to water scarcity in the context of drought can inform interventions for mitigating drought impacts. Considering drought occurrence is expected to increase in the future, intensifying in severity, duration, and impacts, addressing water scarcity requires innovative solutions. Recognition of the socio-economic context, including existing social-economic disparities, psychosocial barriers, and administrative/organisational constraints should be central to crafting intervention strategies. For example, technical solutions are important but insufficient to address wasteful practices and financial challenges. For instance, water production and harvesting infrastructure might be beyond the reach of cash-strapped municipalities and poor households. Poor households often have small houses and yards, and hence low roof surface area for capturing rainwater, limited yard space for installing water tanks, and no expendable income to cover the high costs of installing water harvesting technology (gutters, connection pipes, and tanks). Failure to understand water supply challenges as a complex interplay of physical, historical, and socio-economic factors will have a disproportionate impact on vulnerable groups, which represents a disservice to the fight for equitable access to water. Wasteful water use practices and a lack of investment in water conservation measures can be addressed by combining water infrastructure and

behavioural interventions [41]. Given that in the face of water scarcity, water supply is determined by consumption patterns, behaviour change strategies offer a cheap and long-term pathway to sustainable water use and reliable and sufficient water supply. Without investment in behavioural strategies, urban authorities will struggle to meet the growing demand for water, with a disproportionate impact on vulnerable groups, which can be a source of conflicts. Long-term solutions to water security lie in inclusive and pro-poor approaches. This is especially important given the legacy of unequal access to services in urban spaces in South Africa, and the mandate of municipalities to ensure equitable access to water.

**Author Contributions:** Conceptualisation, G.T. and S.R.; methodology, all authors; validation, all authors; formal analysis, A.P. and G.T.; investigation, A.P.; writing—original draft preparation, G.T.; writing—review and editing, all authors; visualisation, G.T.; supervision, G.T. and S.R.; funding acquisition, G.T. All authors have read and agreed to the published version of the manuscript.

**Funding:** This research was funded by a Rhodes University Research Committee Grant, 2019.

**Institutional Review Board Statement:** The study was conducted according to the guidelines of the Declaration of Helsinki, and approved by the Ethics Review Committee of Rhodes University (Review Reference: 2019-0560-530) on 1 July 2019.

**Informed Consent Statement:** Informed consent was obtained from all subjects involved in the study.

**Data Availability Statement:** The data that support our research findings are available from the corresponding author on request.

**Acknowledgments:** A.P. would like to thank the National Research Fund and JB Marks Education Trust for funding her studies. A.P. would also like to thank field assistants and data capturers, namely Sinako Mtakati, Sanelisiwe Skhakhane, Nadia Schmidtke, Ezile Mdikwa and Ayanda Batyi. Lastly, G.T. thanks Rhodes University for the Rhodes University Research Committee Grant, 2019.

**Conflicts of Interest:** The authors declare no conflict of interest. The funders had no role in the design of the study; in the collection, analyses, or interpretation of data; in the writing of the manuscript, or in the decision to publish the results.

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
