# Peer review of "Persistent Droughts and Water Scarcity: Households’ Perceptions and Practices in Makhanda, South Africa"

_land, doi:10.3390/land10060593_

Round 1

Reviewer 1 Report

Dear authors, 

As per my view, paper is good and written well but i don't know why you have submitted to Land journal of MDPI, better suited for WATER, MDPI. Anyway, kindly revised the introduction as state-of the art is missing. Secondly, discussion part is week, author need to discuss much about the water security an challenges at regional and national level to create interest to the reader widely. Before considered for publication, manuscript need substantial revision as suggested below:  

Abstract and conclusion written generalized, kindly re-write in fruitful way as this paper is very important for the paper. 

Author Response

Thank you for the critical comments. We have addressed these in the revised version as follows:

Comment 1: As per my view, paper is good and written well but i don't know why you have submitted to Land journal of MDPI, better suited for WATER, MDPI.

Response 2: Thank you for the complimentary comment.

Comment 2: Anyway, kindly revised the introduction as state-of the art is missing.

Response 2: We thank the reviewer for this comment, but we were not sure what exactly we should cover in the revisions. We appreciate if there can be clarity on this comment.

Comment 3: Secondly, discussion part is week, author need to discuss much about the water security and challenges at regional and national level to create interest to the reader widely.

Response 3: We have addressed this comment as follows:

The results have practical implications for a reliable availability of sufficient water needed to sustain society. For example, predictions of severe and widespread droughts in the future mean that water security will be a serious service delivery challenge, particularly in contexts where water supply is already problematic [8]. Protecting society from the extremes of drought requires investments in both technical and behavioural strategies for managing water supply in cities. The way water is managed and used will have implications on the supply of water, and in turn, on the social and economic activities needed for human well-being. For example, the local municipality reports failure to supply water to consumers in an equitable way or transfer water to low level reservoirs due to residents’’ failure to use water sparingly and reasonably. For example, when water is restored for a few hours, some residents quickly irrigate gardens and fill up their rain tanks, and swimming pools which means it is not possible to transfer water from one area to the other, illustrating the important role of sustainable water use practices in crafting interventions for water security. Beyond the local level, our results have national implications. At the national level, water insecurity is increasingly a huge challenge with many cities reporting water supply failures. While water supply failure is in part explained by climate change-related events (frequent droughts, heat waves and late onset of rains), equitable use of the little water available will depend on consumers’ use practices. While the water supply problem can vary from place to place, it is plausible to suggest that technical intervention measures supported with behavioural strategies can help mitigate the impacts of droughts. A shift from technical centred interventions to techno-behavioural interventions might ensure supply of reliable and adequate water to residence in the face of an increasingly arid conditions, and their negative impacts on water availability. This shift is transformative in its nature because it can allow equitable sharing of scarce water resources, an aspect that is especially important to consider in contexts characterised by huge inequalities in basic service provision, such as water supply. Interventions that engender sustainable water use practices can allow city authorities and consumers to have a shared understanding of the water supply challenge, and highlight the responsibility of each stakeholder in managing whatever water is available.

Comment 4: Increasing drought under global warming in observations and models. Abstract and conclusion written generalized, kindly re-write in fruitful way as this paper is very important for the paper.

Response 4: We have revised the abstract and conclusions as per the reviewer’s comment.

Reviewer 2 Report

It is an interesting paper that makes a good, empirically sound contribution to knowledge in a very important field of study. I does not require revision. It is well written and presented and, therefore, is suitable for publication. 

Author Response

Thank you for the positive comments about the value of the paper.